# Production and Activity of Cristazarin in the Lichen-Forming Fungus *Cladonia metacorallifera*

**DOI:** 10.3390/jof7080601

**Published:** 2021-07-26

**Authors:** Min-Hye Jeong, Chan-Ho Park, Jung A Kim, Eu Ddeum Choi, Soonok Kim, Jae-Seoun Hur, Sook-Young Park

**Affiliations:** 1Korean Lichen Research Institute, Sunchon National University, Sunchoeon 57922, Korea; minhye1962@gmail.com (M.-H.J.); luckyluky@nate.com (C.-H.P.); 2Department of Plant Medicine, Sunchon National University, Suncheon 57922, Korea; nayaced@korea.kr; 3National Institute of Biological Resources, Incheon 22689, Korea; jakim21@korea.kr (J.A.K.); sokim90@korea.kr (S.K.)

**Keywords:** *Cladonia metacorallifera*, cristazarin, lichen bioresource, polyketide synthase (PKS) genes, secondary metabolites

## Abstract

Lichens are a natural source of bioactive compounds. *Cladonia metacorallifera* var. *reagens* KoLRI002260 is a rare lichen known to produce phenolic compounds, such as rhodocladonic, thamnolic, and didymic acids. However, these metabolites have not been detected in isolated mycobionts. We investigated the effects of six carbon sources on metabolite biosynthesis in the *C. metacorallifera* mycobiont. Red pigments appeared only in Lilly and Barnett’s media with fructose at 15 °C after 3 weeks of culture and decreased after 6 weeks. We purified these red pigments using preparative-scale high performance liquid chromatography and analyzed them via nuclear magnetic resonance. Results indicated that 1% fructose-induced cristazarin and 6-methylcristazarin production under light conditions. In total, 27 out of 30 putative polyketide synthase genes were differentially expressed after 3 weeks of culture, implying that these genes may be required for cristazarin production in *C. metacorallifera.* Moreover, the white collar genes *Cmwc-1* and *Cmwc-2* were highly upregulated at all times under light conditions, indicating a possible correlation between cristazarin production and gene expression. The cancer cell lines AGS, CT26, and B16F1 were sensitive to cristazarin, with IC_50_ values of 18.2, 26.1, and 30.9 μg/mL, respectively, which highlights the value of cristazarin. Overall, our results suggest that 1% fructose under light conditions is required for cristazarin production by *C. metacorallifera* mycobionts, and cristazarin could be a good bioactive compound.

## 1. Introduction

Lichens are symbiotic organisms composed of a lichen-forming fungus (the mycobiont) and an alga, a cyanobacterium, or both (the photobiont). Lichens produce various characteristic secondary metabolites, such as depsides, depsidones, dibenzofurans, pulvinates, chromones, and quinones. Notably, these metabolites, which have been detected in extracts of lichen thalli, are also produced when isolated mycobionts are cultured without their algal partners [1,2,3,4]. However, some mycobiont isolates cannot produce the metabolites that are detected in their symbiotic lichen form [5,6]. While the precise causal factors are unknown, it is expected that differences in nutrient conditions, especially carbon sources, account for changes in the polyketide biosynthesis pathway, which lead to differences in secondary metabolite profiles [7]. The secondary metabolite productions in various lichens with different carbohydrate carbon sources have been compared [8,9,10,11]. Lichen mycobionts have been reported to produce secondary metabolites in culture media supplemented with 10–20% sucrose [9,10,11,12], mannitol, or ribitol [8].

The red compounds cristazarin (3-ethyl-2-hydroxy-7-methoxynaphthazarin) and 6-methylcristazarin are derived from naphthazarin (5,8-dihydroxy-1,4-naphthoquinone; Figure 1) and produced by the isolated *Cladonia cristatella* mycobiont [6]. These compounds have been identified in solid or liquid cultures of the *C. cristatella* mycobiont [6,12], but not in the lichen thalli of *C. cristatella* nor in other lichens to date. Furthermore, cristazarin displays potent biological properties, including antibacterial and antitumor activities [7], and hence may be a potential antibacterial and antitumor agent. The red compounds cristazarin (3-ethyl-2-hydroxy-7-methoxynaphthazarin) and 6-methylcristazarin are derived from naphthazarin (5,8-dihydroxy-1,4-naphthoquinone; Figure 1) and produced by the isolated *Cladonia cristatella* mycobiont [6]. These compounds have been identified in solid or liquid cultures of the *C. cristatella* mycobiont [6,12], but not in the lichen thalli of *C. cristatella* nor in other lichens to date. Furthermore, cristazarin displays potent biological properties, including antibacterial and antitumor activities [7], and hence may be a potential antibacterial and antitumor agent

The majority of secondary metabolites in fungi and lichens are aromatic polyketides, such as cristazarin [13]. Polyketide metabolites constitute a structurally diverse family of natural compounds. They are produced by the successive condensation of acyl coenzyme A (CoA) subunits in a head-to-tail manner via the polyketide pathway [14,15], and they are synthesized by polyketide synthases (PKSs). In recent years, several advanced techniques, including whole-genome sequencing and bioinformatics, have offered new opportunities to identify putative PKS genes via known domains such as the keto-synthase (KS), acyltransferase (AT), and acyl carrier (AC), and to predict the putative chemical structures of their corresponding secondary metabolites without genetic nor chemical analyses [16,17,18,19,20,21,22,23,24]. Moreover, the analysis of gene expression provides a comprehensive understanding of gene regulation under specific conditions.

Light is a major environmental signal that influences various biological activities. In fungi, light is a source of information, such as the induction or inhibition of morphogenesis, reproduction, phototrophy, circadian clock resetting, and secondary metabolism [25,26,27,28,29,30]. Furthermore, the production of a number of secondary metabolites in fungi is induced by light [27,28,30,31,32]. The photoreceptors, including white collar photoreceptors, have been identified and characterized [27,28,30,31,32]. The white collar genes *wc-1* and *wc-2* have been reported to be involved in several biological processes, such as the production of secondary metabolites [27,28,30,31,32]. In lichen, numerous secondary metabolites have been reported [14,33]. However, little is known about the correlation between secondary metabolites and white collar genes in lichens because of limitations in the functional analysis of lichen mycobionts. 

In a previous study, we have sequenced, assembled, and analyzed the genome of the lichen-forming fungus *C. metacorallifera* [34], which is a species closely related to *C. cristatella* [35]. The genome sequence of *C. metacorallifera* contains approximately 11,361 genes across 36.7 Mb, and a total of 30 putative PKS genes that code for KS, AT, and AC domains have been identified via bioinformatics [34]. To date, none of these genes have been analyzed structurally nor functionally, except for the *CmPKS1* gene [36]. Moreover, the lichen thalli of *C. metacorallifera* are known to produce rhodocladonic, thamnolic, and didymic acids. However, these metabolites have not been detected in the isolated *C. metacorallifera* mycobiont.

The objective of this study was to investigate whether the isolated *C. metacorallifera* mycobiont can produce secondary metabolites from different carbon sources under appropriate conditions. We discovered that the *C. metacorallifera* mycobiont produces cristazarin on a medium supplemented with 1% fructose. We demonstrated that fructose and light induced the production of secondary metabolites, namely cristazarin and 6-methylcristazarin, in this fungus. We also highlighted that purified cristazarin suppressed the growth of several cancer cell lines. Thus, the cultured *C. metacorallifera* mycobiont may be a good biological material to reveal the biosynthetic switching mechanism of cristazarin at the molecular level. Finally, we hypothesized that the mycobiont *C. metacorallifera* alone could produce cristazarin rather than some lichen substances without the partner algae *Coccomyxa* sp.

## 2. Materials and Methods

### 2.1. Fungal Isolate

*Cladonia metacorallifera* var. *reagens* KoLRI002260 was obtained from the Korean Lichen and Allied Bioresource Center (KOLABIC) of the Korean Lichen Research Institute (KoLRI) at Sunchon National University, Korea [34].

### 2.2. Media and Culture Conditions

The fungal isolate was grown at 15 °C under dark conditions on malt-yeast (MY) agar medium (i.e., 15 g malt extract broth [Difco] and 15 g/L agar) or Lilly and Barnett’s (LB) liquid medium (i.e., 10 g glucose; 2 g asparagine; 1 g KH_2_PO_4_; 0.5 g MgSO_4_·7H_2_O; 0.2 mg Fe(NO_3_)_3_·9H_2_O; 0.2 mg ZnSO_4_·7H_2_O; 0.1 mg MnSO_4_·4H_2_O; 0.1 mg thiamine; and 5 μg/L biotin) [37]. DNA and RNA were isolated from mycelia, which were grown on a liquid LB medium for 14 days. LB medium without a carbon source was used as the basal medium. Solid media were prepared using 1.5% agar prior to autoclaving. A 1% (*w*/*v*) carbon source, namely glucose, fructose, sucrose, ribitol, mannitol, or sorbitol, was added to the LB medium without glucose and sterilized via autoclaving.

To induce the production of secondary metabolites by the *C. metacorallifera* mycobiont, 2-month-old mycelial plugs (3 mm in diameter) freshly grown on MY agar medium were gently crushed with 1 mL sterilized distilled water using a sterile mortar and a pestle. A total of 100 μL of the homogenized fungal suspension was dropped on a sterilized 0.45 μm pore cellulose nitrate membrane (47 mm diameter; Whatman, Cytiva, Marlborough, MA, USA), which was overlaid on a LB medium without a carbon source or with one of the 6 different carbon sources. The fungus on the plate was grown at 15 °C under fluorescent light.

As for liquid cultures, 30 mL LB medium with fructose was prepared in a 100 mL volumetric flask, and 100 μL of the homogenized fungal suspension was transferred to the medium. The inoculated liquid was incubated at 15 °C under fluorescent light (6500 k, 18 wattage) in an orbital shaker (150 rpm) for 1–6 weeks. All samples were harvested from 3 replicates of 3 biological repeats, immediately frozen in liquid nitrogen, and stored at −80 °C until further processing.

### 2.3. Analysis of Secondary Metabolites Using Thin Layer Chromatography (TLC) and High Performance Liquid Chromatography (HPLC)

Secondary metabolites were analyzed via thin-layer chromatography (TLC) and high-performance liquid chromatography (HPLC). Prior to the TLC analysis, lichen thalli and cultured isolated fungal mycelia were soaked in 1 mL acetone for 5 min, and 50 µL of the concentrated extract was spotted around 2 mm in diameter on Silica gel 60 F_254_ pre-coated plates (Merck KGaA, Darmstadt, Germany) using a microcap several times. Solvent A (toluene:dioxin:acetic acid 180:45:5 [*v*/*v*/*v*]), as per Culberson’s improved standardized method [38], was used in this study. The results were examined and marked in daylight and under UV light at 254 nm and 350 nm. As for the HPLC analysis, lichen thalli and freeze-dried fungal mycelia were soaked in 1 mL acetone overnight. To obtain metabolites from fungal cultures in liquid media, cell-free fluid was collected after removing the mycelium via centrifugation (3000 rpm during 10 min) at room temperature. The suspension was treated with an equal volume of ethyl acetate, and the upper phase of the extract was evaporated and dissolved in 1 mL acetone. Acetone extracts were subjected to HPLC analysis (LC-20A, Shimadzu, Kyoto, Japan) on a reversed-phase column (150 mm × 3.9 mm I.D.; YMC-Pack ODS-A, YMC, Kyoto, Japan) containing a fully endcapped C18 material (particle size: 5 μm; pore size: 12 nm). The injection volume of samples used was 10 µL. Before subsequent injection, elution was performed at a flow rate of 1 mL/min under the following conditions: a column temperature of 40 °C and a solvent system consisting of methanol:water:phosphoric acid (80:20:1 [*v*/*v*/*v*]). Analyses were monitored using a photodiode array detector (SPD-M20A, Shimadzu) with a range of 190–800 nm throughout the HPLC run. The observed peaks were scanned between 190 nm and 400 nm. The sample injection volume was of 10 μL. The standards that were used were obtained acetone extracts from the following sources: rhodocladonic acid (retention time [*t*_R_] = 2.6 min) and didymic acid (*t*_R_ = 10.7 min) isolated from the lichen thallus of *Cladonia macilenta*, thamnolic acid (*t*_R_ = 3.3 min) from the lichen thallus of *Thamnolia vermicularis*, and cristazarin (*t*_R_ = 3.4 min) and 6-methylcristazarin (*t*_R_ = 8.1 min) from liquid cultures of *C*. *cristatella* Tuck.

### 2.4. Purification of Cristazarin Peak (t_R_ = 3.4 min) by Prep-HPLC

To confirm the presence of cristazarin based on the observed HPLC peak (*t*_R_ = 3.4 min), putative cristazarin was purified using preparative HPLC (prep-HPLC). A total of 50 μL acetone extract was injected at a time. Following the purification of the 3.4 min *t*_R_ peak (putative cristazarin), the purified metabolite was concentrated after ethyl acetate extraction and then subjected to a nuclear magnetic resonance (NMR) analysis.

### 2.5. NMR Spectroscopy Analysis

The chemical structures of the isolated compounds were determined by 1- and 2-dimensional NMR spectroscopies. ^1^H and ^13^C NMR spectra were measured in CD_3_OD (Cambridge Isotope Laboratories, Tewksbury, MA, USA) with a Bruker AMX-500 spectrometer (Bruker, Billerica, MA, USA) at 500 MHz for ^1^H NMR spectra and at 125 MHz for ^13^C NMR spectra. Chemical shifts were calculated using tetramethylsilane as an internal standard. ^1^H and ^13^C NMR assignments were supported by ^1^H−^1^H correlation spectroscopy, heteronuclear multiple-quantum coherence, and heteronuclear multiple-bond correlation experiments.

### 2.6. Genome Information and PKS Gene Annotation

Genome sequencing information was obtained from NCBI (accession number AXCT02000000) [34]. The annotation of conserved domains was conducted by scanning proteins using Pfam and NCBI CDD. The PKS and non-ribosomal peptide synthetase genes were identified by scanning the online database SBSPKS [16].

### 2.7. Analysis of Transcript Levels

Quantitative real-time PCR (qRT-PCR) was performed to measure transcript levels. Total RNA samples and first-strand cDNA were prepared as previously described [39]. qRT-PCR was conducted in a Hard-Shell 96-well semi-skirted PCR plate (Bio-Rad, Hercules, CA, USA), using a Chromo4 Real-Time PCR Detector (Bio-Rad). Each well contained 5 μL 2 × SYBR Green RT-PCR Reaction Mix (Bio-Rad), 2 μL cDNA (12.5 ng/μL), and 15 pmol of each primer (Table 1). All the reactions were performed with 3 biological replicates, using 3 combined RNA samples extracted from independent fungal materials. A β-tubulin gene was included in the assays as an internal control for normalization (Table 1). All amplification curves were analyzed with a normalized reporter threshold of 0.25 to obtain the threshold cycle (Ct) values. The comparative ΔΔCt method was used to evaluate the relative quantities of each amplified product in the samples. Fold changes were calculated as 2^–^^ΔΔCt^ _ENREF_53 [40].

### 2.8. 3-(4,5-Dimethylthiazol-2-yl)-2,5-diphenyltetrazolium Bromide (MTT) Assay

The human cancer cell lines AGS(gastric cancer), B16F1(melanoma), RV1(Prostate cancer), A549(lung cancer), U251(glioblastoma), and the mouse line CT26(mouse colon cancer), were cultured in Roswell Park Memorial Institute (RPMI) 1640 medium/Dulbecco’s Modified Eagle Medium (DMEM) (Gen Depot, Barker, TX, USA) supplemented with 10% fetal bovine serum and 1% penicillin-streptomycin solution in an incubator with a humidified 5% CO_2_ atmosphere at 37 °C. HaCaT Cell (Human immortalized keratinocytes) was also used as a normal cell line. Cells were obtained from the Korean Cell Line Bank (Seoul, Korea). The treatment of cristazarin was repeatedly assessed at 8 different concentrations: 0.8, 1.6, 3.1, 6.3, 12.5, 25, 50, and 100 μg/mL. Cells (26,104 cells/well) were seeded in a 96-well plate, grown overnight, and then treated for 48 h. Once the treatment was completed, the cultures were supplemented with MTT. After incubation with MTT at 37 °C, cells were lysed with lysis buffer containing 50% dimethyl sulfoxide and 20% sodium dodecyl sulfate, and absorbance was measured at 570 nm using a microplate reader (VERSAmax, Molecular Devices, San Jose, CA, USA). The percentage of viable cells was calculated using the following formula:$$Cell\;viability\left(\%\right)=\left(OD_{e}/OD_{c}\right)*100$$
where *CV* is the cell viability (%), *OD_e_* is the optical density of the experimental samples, and *OD_c_* is the optical density of the control. The half-maximal inhibitory concentration (IC_50_) values were calculated using SPSS software. All experiments were repeated at least 2 times. 

## 3. Results

### 3.1. Characterization of C. metacorallifera

*C. metacorallifera* formed a fruticose and red-fruited lichen (Figure 2a). In a previous study, we have reported the presence of *C. metacorallifera* in South Korea and identified the fungus via its molecular biological characteristics using an isolated mycobiont (Figure 2b) [34]. The thalli of *C. metacorallifera* contained rhodocladonic, didymic, and thamnolic acids (Figure 2c). There are two chemotypes of *C. metacorallifera*, which are referred to as the varieties *metacorallifera* (i.e., usnic, didymic, and squamatic acids) and *reagens* Asahina (i.e., usnic, didymic, and thamnolic acids) [41]. Based on the chemotype reported herein, *C. metacorallifera* KoLRI002260 was identified as var. *reagens*. However, no noticeable metabolites were detected from the isolated *C. metacorallifera* mycobiont that cultured on MY agar medium.

### 3.2. Effects of the Carbon Source on the Production of Secondary Metabolites 

Many lichen mycobionts have displayed increased levels of secondary metabolites in environments with additional or specific carbon sources [12,42]. The cultured *C. metacorallifera* mycobiont showed a different pigmentation on each of the six media, except for the control, which contained no carbon source (Figure 3a).

The acetone extracts contained various pigments (Figure 3b). Specifically, the extract from the LB medium supplemented with fructose as a carbon source included a red pigment (Figure 3b). The HPLC analysis revealed that the extracts did not display any notable peaks, except for the extracts from the fructose medium. Two peaks appeared at a *t*_R_ of 3.4 min and 8.1 min, respectively (Figure 3c). Based on the UV spectrum and the *t*_R_ [6], the peaks were identified as cristazarin (i.e., 3.4 min) and 6-methylcristazarin (i.e., 8.1 min).

### 3.3. Identification of Secondary Metabolites via NMR

The ^13^C NMR spectra were assigned by comparison with the spectra of cristazarin [6]: C1 (δ 176.99), C2 (155.83), C3 (128.44), C4 (181.97), C4a (105.02), C5 (166.69), C6 (109.19), C7 (158.51), C8 (158.61), C8a (111.70), C9 (17.10), C10 (12.99), and C11 (57.14) positions (Table 2). The ^1^H NMR spectra were identical to those reported for cristazarin (Table 2) [6]. These data clearly indicated that the mycobiont *C. metacorallifera* can produce cristazarin on LB medium supplemented with 1% fructose.

### 3.4. Production of Cristazarin in a Liquid-Based System

To develop optimum conditions for the mass production of cristazarin under a liquid-based system, we introduced *C. metacorallifera* in a liquid LB medium and incubated the culture at four different temperatures, namely 10 °C, 15 °C, 20 °C, and 25 °C, and with or without fructose.

After 4 weeks of growth in an LB liquid culture, the color of the experimental culture with 1% fructose changed from white to red only at 15 °C under light conditions (Figure 4a,b). In contrast, the color of the experimental cultures without fructose did not change (Figure 4a,b). We observed various indeterminate forms of crystal structure on intercellular hyphae, except in the control (Figure 4c,d).

We also assessed whether the additional production of cristazarin was in accordance with the increased growth period. We observed that the red pigments gradually increased (Figure 4e), and the HPLC analysis indicated that the production of cristazarin increased only under light conditions after 3 weeks but then decreased after 6 weeks (Figure 4f). These results indicated that *C. metacorallifera* requires fructose as a carbon source, as well as specific light and temperature conditions, to produce cristazarin.

### 3.5. Expression Analysis of 30 Putative PKS Genes in C. metacorallifera

The expression of putative PKS genes (Table 1; Figure 5a) under conditions that generated the production of cristazarin by *C. metacorallifera* (i.e., liquid LB medium supplemented with 1% fructose, for 1–6 weeks) was analyzed to reveal a possible correlation between the expression of PKS genes in *C. metacorallifera* and cristazarin production.

To select the most stable reference gene under the conditions that were assessed, we evaluated six candidate genes: the β-tubulin, α-tubulin, elongation factor 1β, ubiquitin extension protein, actin-2, and glyceraldehyde-3-phosphate dehydrogenase genes. Using the GeNorm software [43], we evaluated which genes were the most stable. As β-tubulin gene transcripts were present at a similar intensity across all the conditions that were evaluated (data not shown), we used the expression of β-tubulin gene as a reference gene and the expression of *C. metacorallifera* under non-treated fructose medium as a control.

The results indicated that most of the PKS genes were differentially expressed under induction medium conditions for 1–6 weeks (Figure 5b), except for three *CmPKS* genes (i.e., *CmPKS8*, *CmPKS24,* and *CmPKS25*). A total of 25 genes were classified into three groups according to their gene expression pattern (Figure 5b). Among these groups, group I-2, which included *CmPKS3*, *CmPKS4*, *CmPKS9,* and *CmPKS15*, was highly upregulated after 3 weeks of culture. This implies that PKS genes may be required for the production of cristazarin in *C. metacorallifera*. In addition, group II was also highly upregulated from weeks 1 to 3, which suggests that these genes may be required for the production of cristazarin at a relatively early stage.

### 3.6. Expression Analysis of White Collar-1 and White Collar-2 in C. metacorallifera

Previous studies have revealed that the transcription factors white collar-1 (*wc-1*) and white collar-2 (*wc-2*) genes are essential for most of the light-mediated processes in several fungal species [27,28,29,30,31,32]. We observed that the production of cristazarin in light differed markedly from that under dark conditions (Figure 4e,f). That is, the mycobiont of *C. metacorallifera* needed time and light to produce cristazarin. Furthermore, *wc-1* and *wc-2* orthologs in *C. metacorallifera* were identified according to the InterPro classification [44] and named *Cmwc-1* and *Cmwc-2*.

To assess the potential correlation between the expression of the white collar genes and the production of cristazarin in *C. metacorallifera*, we measured the expression of the *Cmwc-1* and *Cmwc-2* genes under cristazarin-induced conditions during weeks 1 to 6. Cristazarin was the most abundant after 5 weeks of culture, and it gradually decreased thereafter (Figure 4f). These results indicated that both *Cmwc-1* and *Cmwc-2* genes may be involved in the production of cristazarin in *C. metacorallifera*.

### 3.7. Biological Activity of Cristazarin Produced by C. metacorallifera

In order to investigate the biological activity of cristazarin produced by C. metacorallifera, we determined the cytotoxicity of cristazarin in six cancer cell lines and one non-cancer cell line, namely HaCaT (human keratinocytes), by exposing these cells to different concentrations of cristazarin (concentration range: 0.78–100 μg/mL). The IC_50_ of cristazarin ranged from 18.17 μg/mL to >100 μg/mL in the six cancer cell lines (Figure 6). The AGS, CT26, and B16F1 cell lines were sensitive to cristazarin, with IC_50_ values of 18.2 μg/mL, 26.1 μg/mL, and 30.9 μg/mL, respectively. RV1 and A594 cell lines were less sensitive, with IC_50_ values of 48.3 μg/mL and 55.5 μg/mL, respectively. Finally, the U251 cell line was the most resistant (i.e., IC_50_ = 141.42 μg/mL).

## 4. Discussion

Lichen photobionts are green algae or cyanobacteria that produce different types of carbohydrates via photosynthesis. Green algae produce polyols, such as ribitol in *Trebouxia* spp., *Myrmecia* spp., and *Coccomyxa* spp., sorbitol in *Hyalococcus* spp., and erythritol in *Trentepohlia* spp. [45]. Cyanobacteria produce glucose in *Nostoc* spp., *Scytonema* spp., and *Rhizonema* spp. [46,47]. The algal photobiont of *C. metacorallifera* is *Coccomyxa viridis*, which produces ribitol. Ribitol is a crystalline sugar alcohol (C_5_H_12_O_5_) that is formed by the reduction of ribose and converted into mannitol by the mycobiont via the pentose phosphate pathway [48,49,50]. Converted carbohydrates are rapidly and irreversibly metabolized into various forms and consumed for growth, reproduction, and other metabolic activities or diverted into secondary metabolites [51]. In natural habitats, *C. metacorallifera* produces unique phenolic substances such as didymic, squamatic, barbatic, rhodocladonic, and usnic acids, which are synthesized via the polyketide pathway by an acetyl-CoA carboxylase that produces malonyl-CoA [14,52]. 

According to Yamamoto, Matsubara, Kinoshita, Kinoshita, Koyama, Takahashi, Ahmadjiam, Kurokawa and Yoshimura [6], cristazarin and 6-methylcristazarin are produced by *C. cristatella* mycobionts in MY extract medium without an additional carbon source [6]. However, in our study, a red pigment was produced by *C. metacorallifera* mycobionts only in the fructose-supplemented medium under fluorescent light conditions, and this substance was revealed to be composed of cristazarin and 6-methylcristazarin. It is reported that some lichen substances have been detected only in symbiosis, but others have produced their compounds cultured only in cultured mycobionts [6]. Interestingly, cristazarin has not been detected in lichen thalli to date. Thus, it is necessary to understand the conversion of the carbon source, based on photobiont carbohydrates, to determine secondary metabolite production by mycobionts.

The mycobiont of *C. metacorallifera* produced cristazarin only in the medium with 1% fructose, but not in the media supplemented with glucose, sucrose, ribitol, mannitol, or sorbitol. Specifically, unlike the three kinds of sugar alcohols, pale red pigments were observed in media supplemented with glucose and sucrose. These sugar alcohols and glucose are commonly present in the thalli of lichens that have green algae as photobionts. Thus, sugar alcohols and glucose, which are already familiar with symbiotic mycobionts, can be difficult to become stressors or inducers that express genes related to secondary metabolites. The secondary metabolite production of lichen mycobionts is influenced by the carbon source [8,51]. Conversely, fructose is a monosaccharide that is not produced by symbiotic algae in lichens. Regardless of the production of cristazarin, the most important compounds for the classification of carbohydrates depending on the presence or absence of red pigments are monosaccharides and sugar alcohols. Among the three carbohydrates that produce red pigments, the second classification factor is the difference in carbohydrate structure. Structurally, glucose is an aldose and fructose is a ketose. 

One of the main differences between aldoses and ketoses is that the carbonyl group in aldoses is present at the end of the carbon chain, while that in ketoses is present in the middle of the carbon chain. Moreover, the three sugar alcohols are also derived from the reduction of an aldose (sorbitol and mannitol can be reduced in both aldose and ketose). Among the carbohydrates used in the present study, fructose was the only ketohexose with six carbon sources and the only carbon source that stimulated the production of cristazarin. Based on these results, it is expected that cristazarin will also be produced from sucrose, which is a mixture of fructose and glucose, and actually pale red pigment was shown. A previous study on *C. cristatella* has also highlighted that sucrose promotes pigment production more than sugar alcohols do, and 4% is the optimal concentration in an LB medium [12]. 

Lichens produce many phenolic compounds, such as depsides, depsidones, dibenzofurans, pulvinates, chromones, and quinones. Furthermore, without symbiotic photobionts, mycobionts produce anthraquinones and cristazarin [2,6]. Some carbohydrates are converted via polyketide pathways to form diverse polyketides, which are produced by the successive condensation of acetyl-CoA with malonyl-CoA by PKS [36,52]. The most abundant classes of phenolic compounds, namely depsides and depsidones, are composed of orcinol (orsellinic and resorcyclic acids) or β-orcinol (β-orsellinic acid) type subunits [52,53]. Presumably, the mechanism by which red pigments are produced in a medium containing glucose, fructose, and sucrose may be described via the example of Seliwanoff’s test, which distinguishes between aldose and ketose sugars. The reagents in this test consist of resorcinol and concentrated hydrochloric acid and rely on the principle that aldose and ketose generate light pink and deep red colors, respectively, through acid hydrolysis and resorcinol condensation reactions [54]. Similarly, in our study, the red pigments, or cristazarin, can be produced by the reaction of furfural with orcinol subunits biosynthesized via the PKS pathway. Consequently, glucose (i.e., aldose) and sucrose (i.e., aldose and ketose) may generate a pink color lighter than that produced with fructose.

Many lichen compounds are derived from fatty acids. Acetyl-CoA carboxylase assembles structurally diverse products from simple acyl-CoA substrates via a catalytic cycle that involves reduction and dehydration [52,55]. Kozaki and Sasaki [56] have assumed that, under redox regulation, acetyl-CoA carboxylase is activated by light, and this is consistent with the formation of cristazarin under light conditions that was observed in our study. Therefore, it is presumed that fructose, which is a ketohexose carbohydrate, acts as a substrate via the same pathway and contributes to the production of diverse and specific secondary metabolites such as cristazarin. In the PKS gene analysis, *CmPKS3*, *CmPKS4*, *CmPKS9*, and *CmPKS15*, which belong to group I-2, were highly expressed. This group presumably includes the PKS domain that synthesizes single aromatic ring polyketides, such as orcinol (orsellinic acid). In addition to group I-2, most of the other gene groups were also expressed after 3 weeks of culture, a time which corresponds with the appearance of red pigments. Furthermore, MTT assay of cristazarin clearly showed the potential anticancer drug (Figure 6).

## 5. Conclusions

Overall, we showed that *C. metacorallifera* mycobionts produce special metabolites in media supplemented with 1% fructose. Thus, cultured *C. metacorallifera* mycobionts might represent a good biological material to assess the biosynthetic switching mechanism of cristazarin at the molecular level. As cristazarin has shown antibacterial activity against *Bacillus subtilis* (10 μg/mL) and *Staphylococcus aureus* (20 μg/mL), as well as antitumor activity (100 μM) [7], it could be a promising candidate for the development of antitumor and antibacterial agents. Therefore, further research on cell metastasis using cristazarin should be conducted.

## Figures and Tables

**Figure 1 jof-07-00601-f001:**
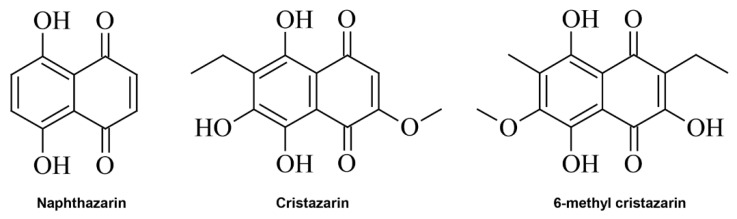
Chemical structure of naphthazarin, cristazarin, and 6-methylcristazarin.

**Figure 2 jof-07-00601-f002:**
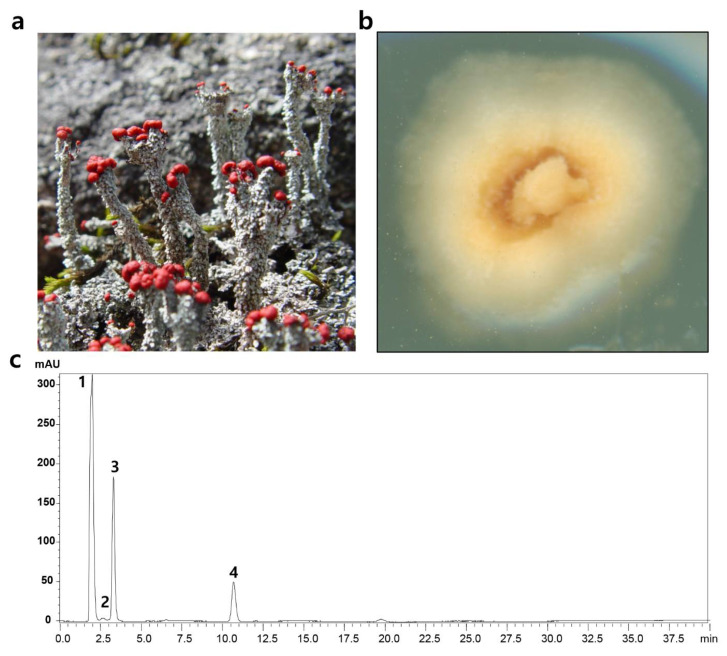
Characteristics of the lichen *Cladonia metacorallifera*. (**a**) Morphology of *C. metacorallifera*; (**b**) isolated *C. metacorallifera* mycobiont; (**c**) high performance liquid chromatography (HPLC) analysis of acetone extracts from *C. metacorallifera* thalli. Compounds were eluted with Me-OH during 40 min and monitored at 254 nm. 1 (retention time [*t*_R_] = 2 min): acetone; 2 (*t*_R_ = 2.6 min): rhodocladonic acid; 3 (*t*_R_ = 3.3 min): thamnolic acid; and 4 (*t*_R_ = 10.7 min): didymic acid.

**Figure 3 jof-07-00601-f003:**
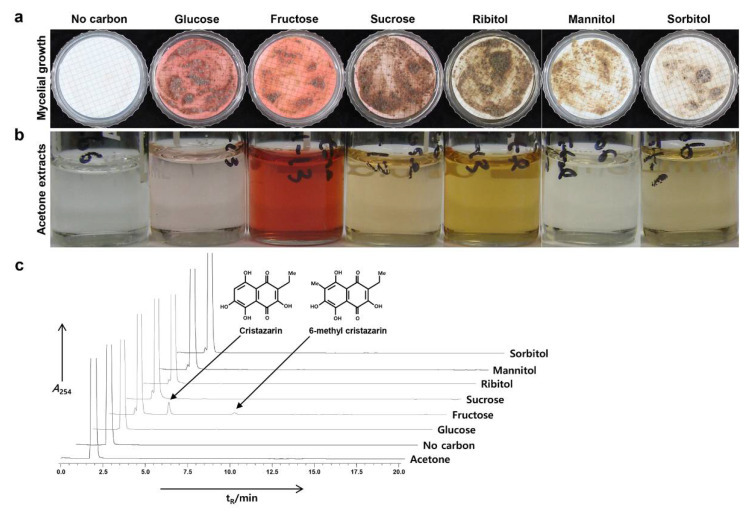
Characteristics of the mycobiont of *C. metacorallifera*. (**a**) Growth of isolated *C. metacorallifera* mycobionts on Lilly and Barnett’s (LB) media without a carbon source or with one of six different carbon sources after three months; (**b**) acetone extracts from the grown mycelia under each carbon condition; (**c**) HPLC analysis of the acetone extracts. Compounds were eluted with Me-OH for 40 min, monitored at 254 nm. Acetone was eluted at *t*_R_ = 2 min as a blank control. 5 (*t*_R_ = 3.4 min): cristazarin; and 6 (*t*_R_ = 8.1 min): 6-methylcristazarin.

**Figure 4 jof-07-00601-f004:**
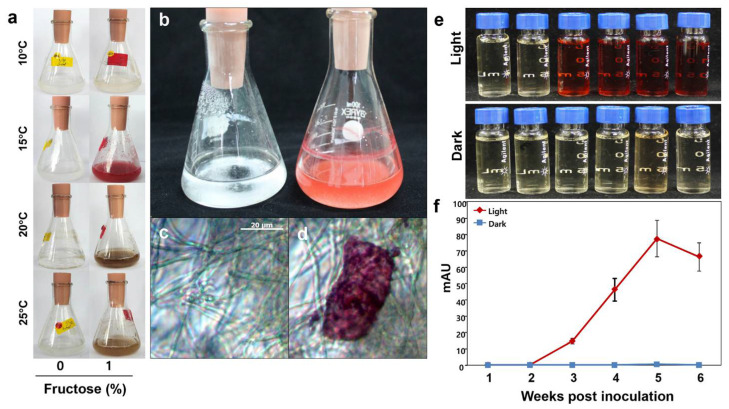
*C. metacorallifera* in liquid culture, and detection of its production of cristazarin 1–6 weeks after inoculation. (**a**) *C. metacorallifera* after 2 months of cultivation in liquid LB media without fructose or with 1% fructose at different temperatures; (**b**) liquid LB cultures of *C. metacorallifera* without (**left**; control) and with (**right**) 1% fructose at 15 °C; (**c**) microscopic observation of the control displayed in Figure 4b; (**d**) microscopic observation of the culture with fructose displayed in Figure 4b, with the presence of a red indeterminate crystal; (**e**) acetone extraction of total substance from *C. metacorallifera* in liquid induction media under light or dark conditions; (**f**) production of cristazarin by *C. metacorallifera* under light or dark conditions (HPLC analysis).

**Figure 5 jof-07-00601-f005:**
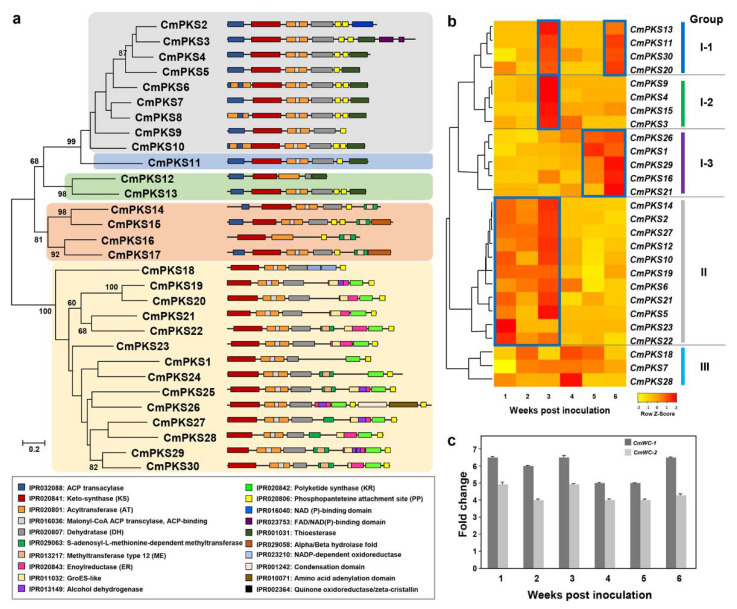
Domain structure and expression profiles of polyketide synthase (PKS) genes; (**a**) the protein domain structures of 30 putative PKS genes were obtained from the InterPro protein database (http://www.ebi.ac.kr/interpro, accessed on 20 July 2021); (**b**) heat map of the expression patterns of 27 PKS genes; (**c**) quantitative RT-PCR analysis of the transcripts from the genes *Cmwc-1* and *Cmwc-2* of *C. metacorallifera* cultured in liquid LB media with 1% fructose during 1–6 weeks.

**Figure 6 jof-07-00601-f006:**
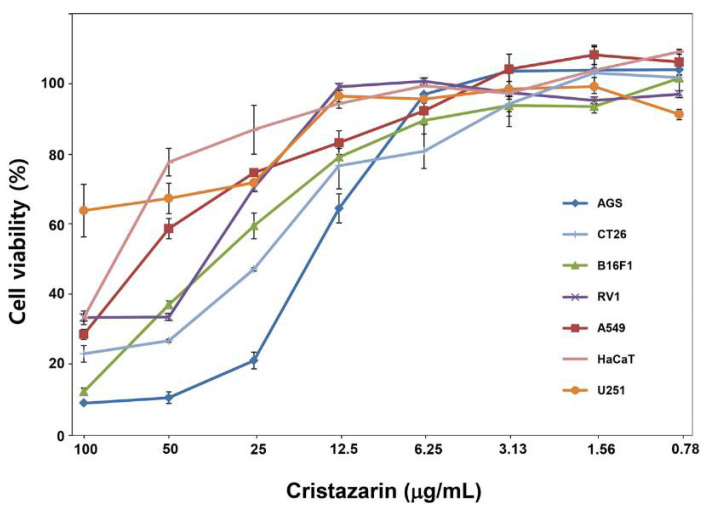
Cytotoxic activity of cristazarin in six different cancer cell lines (AGS, CT26, B16F1, RV1, A549, and U251) and one normal cell line (HaCaT). Cells were counted 48 h after exposure to cristazarin. IC_50_ values were calculated using SPSS. All treatments were triplicated; standard errors are showed.

**Table 1 jof-07-00601-t001:** Primers used for quantitative real-time PCR.

Target Genes *	Forward Primer (5’ → 3’)	Reverse Primer (5’ → 3’)
*β-tubulin*	TGAGGCACTTTACGACATCTG	GAAGTGAAGACGAGGGAAAGG
*α-tubulin*	TCGTCTCTTCAATCACTGCC	GAATTGGATTCATGTGCAGCC
*EF1β*	TCTACAGGGCAAGTTCAATCG	GCAAAGTAGAGGATCAGGAGTG
*UEP1*	CCTCCATCGCATTACCCTTAG	CTTCCCTGCGAAAATCAAACG
Actin2	CACAGCAACCCTATACTCCTTC	CTATTCTCACTATCACACCATCCC
*GAPDH*	AAGACGCTGAATGGGATATGG	TCTTATGCAAGCTTAGCCCTC
*CmPKS1*	GCTGTTTTTGCGGGCATGGA	CATACGGACGGCTTGATGT
*CmPKS2*	ACCAGTTCGGATCACTTC	CGTAGCAATATCTGTTCG
*CmPKS3*	CAAGGCTTCCACTTCTCA	CGGAAGATGTGTAACCTC
*CmPKS4*	TGCGATTCACGCTCCATA	TGACATATCTCTGGCACG
*CmPKS5*	GTCGAACGTATCATTCAT	GATCGATATGAGTGTGCA
*CmPKS6*	TCGAAGAGGTGCAAGCAA	GTTGACCAGCTGTCGAAG
*CmPKS7*	TATCGAGTACACCAGTGC	GCACAGTGTTCTGATCGT
*CmPKS8*	GTCTCATTAGCTATGTAC	ACGGACCAGCTCTCTTGG
*CmPKS9*	AGAGTGCAGCAGAGCTGT	CGTAGTGTCTCTTGGTGC
*CmPKS10*	GCTCCGCAAGAACAGCAA	TGGAATGCGGCAGTGATC
*CmPKS11*	ACTGATGCTTCATGTCGA	TGAGCCGTTGCACCAACA
*CmPKS12*	CAGGTCTTGGAGACTACT	GGTCGACATTCCTGTCTT
*CmPKS13*	ACTAGACAGGCTCCAGAG	TTCATTGTGTCGAAGGTC
*CmPKS14*	TCAGCATACTAACACTGC	TCAGACACCTGAAGACCT
*CmPKS15*	AGGTACATAATCCAGAGA	AGTATACCATTCGCATCG
*CmPKS16*	ATTGGCCTCTTGAGCGCT	GATGTACGTATCCGGATA
*CmPKS17*	ATGGCTGAAGAGGCGGAT	GCGCGACCAATTGAATCA
*CmPKS18*	ATGGAGATGGCGATACGA	TGAGAGTACCAGCCGCAT
*CmPKS19*	TGTGGATGTTGCCTGTCA	GACAGCATCTTCGAGACG
*CmPKS20*	GAGGAGAAGTGGCATCTA	GCATTCTCAAGTCCTTCA
*CmPKS21*	TGGCTTCTGATTATACCACG	GCTAACGTTCGGAGACGATG
*CmPKS22*	ATGGAGCTCTGCATGGTA	TCATCGGCATTGTGAATC
*CmPKS23*	CAGCATACCTGCCGAGAG	ACACCGCATTCTCAAGT
*CmPKS24*	ACACATAATGAAGACATC	GTCGATAATGTTCTGGAG
*CmPKS25*	TCATCGGCACAGTGCACA	GTATAGCAATGCGATATC
*CmPKS26*	ACATCGTGGAGCAGAAGG	TGTGATGCCAGCGTCTTCT
*CmPKS27*	CACAGTGGTACGAAGACA	CGATGTAGCATGAGGTAT
*CmPKS28*	GCGACGTAGATGGATATG	GATGATTGGTTGCTGGAC
*CmPKS29*	GGCGAGACGATATTGATCCAT	CTTGGCCAGTTGAACCG
*CmPKS30*	TGTTAGACAAGCTCACTT	TGAATATGATGCTATCGT
*Cmwc1*	TGCTAATTGCCATACCCG	CCGTAGCTGAATTGTGTGAG
*Cmwc2*	TCGCTTCTTCTACCGCTT	CTTGGTTAGGCGCTCATT

** EF1β:* elongation factor1-*β*; *UEP1*: ubiquitin extension protein gene; GAPDA: glyceraldehydes-3-phosphate dehydrogenase.

**Table 2 jof-07-00601-t002:** NMR data.

C No.	^13^C ^a^	^1^H (Number of Protons, Multiplicity) ^b^ (j-Hz) ^c^
1	176.99	
2	155.83	
3	128.74	
4	181.97	
4a	105.02	
5	166.69	
6	109.19	6.62 (1H, s)
7	158.51	
8	158.61	
8a	111.70	
9	17.10	2.61 (2H, q, 7.5)
10	12.99	1.11 (3H, t, 7.5)
11	57.14	3.93 (3H, s)

^a^ 125 MHz.; ^b^ multiplet; q: quadruplet, s: singlet, t: triplet; ^c^ 500 MHz.

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
