# Peer review of "Production and Activity of Cristazarin in the Lichen-Forming Fungus Cladonia metacorallifera"

_jof, 2021, doi:10.3390/jof7080601_

Round 1

Reviewer 1 Report

 This is a manuscript revision for the article 'Production and activity of cristazarin in the lichen-forming fungus Cladonia metacorallifera'

The manuscript is well written and organized, However, the issues below should be considered by the authors to improve the manuscript.

Figure 1. What is the source? Provide a reference for these figures.

Table 1 legend. Include more information

Lines 54-55: Authors state Furthermore, cristazarin displays potent biological properties, including antibacterial and antitumor activities [7]. Reference 7 is 23 years old. Is there anything more recent? Ref 6 is also by the same author and 25 years old.

Line 130: What are the make and manufacturers of the fluorescent light used?  It is a key part of the study, so more information like watt or operating power inscribed on the fluorescent light should be provided.

Line 138: ........... WHITE COLLAR-1 and WHITE COLLAR-2 - ................ Why are these in capitals?

Lines 192-193: All the reactions were performed with more than two biological replicates...... How many replicates exactly?

Line 200: Define MTT in full the first time.

Line 201: How was the human cancer cell obtained? If commercial state company and city.

Line 217: Provide a reference for the equation used

Line 217: What is 6100?

Line 216: Delete Table 1. Primers used for quantitative real-time PCR.

Lines 227-229: Authors state However, we did not detect any metabolites from the isolated C. metacorallifera mycobiont cultured on MY agar medium. Do you mean you did not detect the metabolites investigated or metabolites within the threshold set? When microorganisms are grown in a media, there must be some sort of metabolite produced. Rephrase.

Line 262: More words are needed in the NMR data legend. Define q, t etc. Explain what the table is showing.

Figure 6: Why was an antimicrobial activity not monitored between 0 and 48 hours?

Lines 425-437 appear to be the conclusion and it was based on the anti-tumor and antibacterial activities proposed in reference 7. Again what has happened between reference 7 and today? Have other investigators not done any work? Authors should highlight if the range of effectiveness or antibacterial property found with the compounds studied is similar to other known agents from investigations by other workers.

What are the limitations of this study?

Author Response

Reviewer 1.

  1. Figure 1. What is the source? Provide a reference for these figures.

-- We added reference in line number 113.

  1. Table 1 legend. Include more information.

-- We added more information in Table 1.

  1. Lines 54-55: Authors state Furthermore, cristazarin displays potent biological properties, including antibacterial and antitumor activities [7]. Reference 7 is 23 years old. Is there anything more recent? Ref 6 is also by the same author and 25 years old.

-- There are no more bioactivity experiments about the cristazarin. In 2014, a report of mass culture was published for clothes dyeing by Dr. Yamamoto. However, this report does not correspond to bioactivity.

  1. Line 130: What are the make and manufacturers of the fluorescent light used?  It is a key part of the study, so more information like watt or operating power inscribed on the fluorescent light should be provided.

-- We added the information.

  1. Line 138: ........... WHITE COLLAR-1 and WHITE COLLAR-2 - ................ Why are these in capitals?

-- We corrected the information.

  1. Lines 192-193: All the reactions were performed with more than two biological replicates...... How many replicates exactly?

-- We corrected the information. All the reactions were performed with three biological replicates.

  1. Line 200: Define MTT in full the first time.

-- 2.8. title was edited to 3-(4,5-dimethylthiazol-2-yl)-2,5-diphenyltetrazolium bromide (MTT) assay.

  1. Line 201: How was the human cancer cell obtained? If commercial state company and city.

-- We added the information.

  1. Line 217: Provide a reference for the equation used

Line 217: What is 6100?

-- We corrected the information.

  1. Line 216: Delete Table 1. Primers used for quantitative real-time PCR.

- We deleted.

  1. Lines 227-229: Authors state However, we did not detect any metabolites from the isolated C. metacorallifera mycobiont cultured on MY agar medium. Do you mean you did not detect the metabolites investigated or metabolites within the threshold set? When microorganisms are grown in a media, there must be some sort of metabolite produced. Rephrase.

- We edited.

  1. Line 262: More words are needed in the NMR data legend. Define q, t etc. Explain what the table is showing.

- We edited the table legend.

  1. Figure 6: Why was an antimicrobial activity not monitored between 0 and 48 hours?

-- MTT assay was performed for exam of anticancer bioactivity from cell viability not antimicrobial activity. Almost MTT test observe the samples of 48 hours after treatments.

  1. Lines 425-437 appear to be the conclusion and it was based on the anti-tumor and antibacterial activities proposed in reference 7. Again what has happened between reference 7 and today? Have other investigators not done any work? Authors should highlight if the range of effectiveness or antibacterial property found with the compounds studied is similar to other known agents from investigations by other workers

-- We don’t know exactly what happen during two decades. However, we did not find any meaningful papers so far. Again, we can say there are no more bioactivity experiments about the cristazarin.

  1. Some lichen substances like usnic acid, cristazarin, protolichesterinic acid, polyporic acid, depsidone and lichenin have been investigated for antitumor effects on tumorcells—melanoma B-16 (Khanuja et al., 2007), P388 leukaemia (Takai et al., 1979), K-562 leukaemia (Hirayama et al.,1980), Ehrlich solid tumor (Cain, 1966) and lymphocyte(Correche et al., 2002) cells. In vitro anticancer activities of lichen extracts have been evaluated according to the cellproliferation assay (Tokiwano et al., 2009) in three cancercell lines: human pancreatic (PANC-1) (Ingolfsdottir et al.,2002), prostate (DU-145) (Russo et al., 2006) and breast(MCF7) (Bogo et al., 2010) cancer cell lines. What are the limitations of this study?

-- In this study, we developed production of cristazarin in C. metacorallifera with diverse carbon sources and focused on the development for their use. Basically, I did not fully understand the upper questions.

Reviewer 2 Report

I have some comments to improve the quality of manuscript entitled "Production and activity of cristazarin in the lichen-forming fungus Cladonia metacorallifera". In general, it is well-written and -conducted, the discussion needs to be improved, and conclusion does not exist. Please find my suggestions and comments below:

Introduction

In my opinion the bioactivities of naphthazarin, cristazarin, and 6-methylcristazarin should be discussed more

In Fig. 1, please draw the structures by a software e.g. ChemDraw, their qualities are not acceptable

L86-88: please reword "However, ... mycobionts."

L98-108: at the end of Introduction, the aim of the study should be mentioned, however in the present form the general outcomes are reported.

Materials & methods

L120: please use "day(s)" instead of "d" in "14 d", or define before use

L142-153: please determine concentrations of the extracts applied in the TLC and HPLC analysis.

L160-165: please describe how the standards have been prepared.

L200: please explain the cell lines experimented are belonged to which cancer types? how about positive control(s)? 

Results

I think the HPLC method could be optimized, all four compounds are eluted within 10 min, while the method duration is 40 min!

Discussion

L360-361: please cite the reference according to the journal's guidelines

L366: authors: "Interestingly, cristazarin has never been detected in lichen thalli to date", what would be the rationale? please shortly discuss.

- the cytotoxicity of the compounds has not been discussed. 

there is no Conclusion section, please add this part.

Reference

in some refs. there is no doi numbers, please add,

furthermore, the journal's names have to be in abbreviated form, please check all

Author Response

Introduction

  1. In my opinion the bioactivities of naphthazarin, cristazarin, and 6-methylcristazarin should be discussed more.

-- In this study, we focused on production of cristazarin in C. metacorallifera with diverse carbon sources and also focused on the development for their use, but not the chemistry about bioactivity of naphthazarin, cristazarin, and 6-methylcristazarin. This study will be one of good research topic in the future.

  1. In Fig. 1, please draw the structures by a software e.g. ChemDraw, their qualities are not acceptable

-- We replaced the fig. 1 that generated by a software ChemDraw.

  1. L86-88: please reword "However, ... mycobionts."

-- We corrected.

  1. L98-108: at the end of Introduction, the aim of the study should be mentioned, however in the present form the general outcomes are reported.

-- We corrected.

Materials & methods

  1. L120: please use "day(s)" instead of "d" in "14 d", or define before use

-- We corrected the “d” to “days”.

  1. L142-153: please determine concentrations of the extracts applied in the TLC and HPLC analysis.

-- We added the information.

  1. L160-165: please describe how the standards have been prepared.

-- We added the information.

  1. L200: please explain the cell lines experimented are belonged to which cancer types? how about positive control(s)? 

-- We added the information.

Results

  1. I think the HPLC method could be optimized, all four compounds are eluted within 10 min, while the method duration is 40 min!

-- We used a method that usually takes about 40 minute to analyze using the existed lichen metabolite library.

Discussion

  1. L360-361: please cite the reference according to the journal's guidelines

-- We corrected.

  1. L366: authors: "Interestingly, cristazarin has never been detected in lichen thalli to date", what would be the rationale? please shortly discuss.

-- We added the information.

  1. the cytotoxicity of the compounds has not been discussed. 

-- In the last part of discussion, we mentioned that MTT assay of cristazarin showed the potential anticancer drug.

  1. there is no Conclusion section, please add this part.

 -- The last part of discussion would be [conclusion] section.

Reference

  1. in some refs. there is no doi numbers, please add, furthermore, the journal's names have to be in abbreviated form, please check all

-- We corrected.

Round 2

Reviewer 2 Report

Dear Authors,

Thank you for the revision, unfortunately there are still two issues concerning the following:

the remark: the cytotoxicity of the compounds has not been discussed. 

reply: In the last part of discussion, we mentioned that MTT assay of cristazarin showed the potential anticancer drug.

I can not see any discussion regarding cytotoxicity in the "Discussion" section.

the remark: there is no Conclusion section, please add this part.

reply: The last part of discussion would be [conclusion] section.

please write a heading "Conclusion" to easier following up the paper.

Moreover, the references' font is not in style of the journal

Author Response

  1. the remark: the cytotoxicity of the compounds has not been discussed. 

reply: In the last part of discussion, we mentioned that MTT assay of cristazarin showed the potential anticancer drug.

I can not see any discussion regarding cytotoxicity in the "Discussion" section.

-- We added the sentence in Discussion.

  1. the remark: there is no Conclusion section, please add this part.

reply: The last part of discussion would be [conclusion] section.

please write a heading "Conclusion" to easier following up the paper.

-- We added “5. Conclusions”.

  1. Moreover, the references' font is not in style of the journal

-- We corrected the reference’s font.